# Prenatal exposure to per- and polyfluoroalkyl substances (PFAS) and incidence of asthma and wheeze in childhood: A register-based cohort study in Ronneby, Sweden

Annelise J. Blomberg[1]*, Christel Nielsen[1,2], Beata Borgström Bolmsjö[3,4], Marie-Abèle Bind[5,6], Linda Hartman[7], Anna Saxne Jöud[1,8]

**1** Division of Occupational and Environmental Medicine, Department of Laboratory Medicine, Lund University, Lund, Sweden, **2** Department of Clinical Pharmacology, Pharmacy and Environmental Medicine, Institute of Public Health, University of Southern Denmark, Odense, Denmark, **3** Center for Primary Health Care Research, Department of Clinical Sciences Malmö, Lund University, Malmö, Sweden, **4** University Clinic Primary Care Skåne, Region Skåne, Sweden, **5** Biostatistics Center, Massachusetts General Hospital, Boston, Massachusetts, United States of America, **6** Department of Medicine, Harvard Medical School, Boston, Massachusetts, United States of America, **7** Division of Mathematical Statistics, Lund University, Lund, Sweden, **8** Division of Orthopaedics, Department of Clinical Sciences Lund, Lund University, Lund, Sweden

* annelise.blomberg@med.lu.se

## Abstract

### Background

Early-life exposure to per- and polyfluoroalkyl substances (PFAS) may impact the developing lungs and immune system and increase the risk of childhood asthma, but no studies have been conducted in a high-exposed population. The objective of this study was to estimate associations between prenatal PFAS exposure and childhood incidence of asthma and wheeze in Blekinge County, Sweden, where a subset of residents in the city of Ronneby was exposed to PFAS from drinking water contaminated by aqueous film-forming foam (AFFF).

### Methods and findings

We constructed a register-based open cohort of 11,488 children born in Blekinge county between 2006 and 2013 and followed each individual from birth until age 12 or December 31, 2022. Maternal address history was linked to water distribution records to create a categorical proxy variable for prenatal PFAS exposure from drinking water. We identified incident cases of wheeze and asthma from administrative health records and estimated hazard ratios (HRs) using Cox proportional hazards models adjusted for individual-level confounders, including maternal smoking in early pregnancy, maternal age at delivery, parity, child sex, parental asthma, and socioeconomic factors. As a secondary analysis, we applied a Rubin Causal

which permits unrestricted use, distribution, and reproduction in any medium, provided the original author and source are credited.

**Data availability statement:** Data cannot be shared publicly because they contain sensitive personal information and are subject to legal and ethical restrictions under Swedish law. Access to the data is restricted to research projects approved by the Swedish Ethical Review Authority and the relevant registry holders; for some registers, access is limited to researchers affiliated with Swedish institutions. Applications for data access must be submitted directly to the respective registry owners. Further details on data access procedures are available at the websites provided below. Swedish National Board of Health and Welfare (https://bestalladata.socialstyrelsen.se/data-for-forskning/): National Patient Register, National Prescribed Drug Register, and National Medical Birth Register. Statistics Sweden (https://www.scb.se/vara-tjanster/bestall-data-och-statistik/mikrodata/): Total Population Register and the Longitudinal Integrated Database for Health Insurance and Labor Market Studies (LISA). Region Blekinge (https://regionblekinge.se/halsa-och-vard/forskning-och-utveckling.html): Blekinge Healthcare Register. The analysis code and results are openly available in a published GitHub repository (https://github.com/ajblomberg/PFAS-and-Childhood-Asthma) that has been permanently archived on Zenodo with a DOI (https://doi.org/10.5281/zenodo.17931667).

**Funding:** This work was funded by the European Union's Horizon Europe program under the Marie Skłodowska-Curie Postdoctoral Fellowships (https://marie-sklodowska-cu-rie-actions.ec.europa.eu/actions/postdoctor-al-fellowships; grant number 101058697 to AJB) and the Swedish Research Council for Health, Working Life and Welfare (FORTE, https://forte.se/en; grant number 2020-00112 to ASJ and 2024-00748 to AJB). The funders had no role in study design, data collection and analysis, decision to publish, or preparation of the manuscript.

**Competing interests:** The authors have declared that no competing interests exist.

**Abbreviations:** AFFF, aqueous film-forming foam; EDCs, endocrine-disrupting chemicals; DAG, directed acyclic graph; HRs, hazard ratios; IQR, interquartile range; MICE, multiple

Model (RCM) analysis to estimate the average marginal effect of prenatal PFAS exposure on wheeze and asthma among the very highly-exposed population, using a matched dataset of very-high and background-exposed individuals balanced on measured confounders. Overall, 18% of children were diagnosed with wheeze and 17% with asthma during follow-up. Very high prenatal PFAS exposure was associated with incidence of asthma (HR: 1.44, 95% CI [1.08, 1.92]), whereas no associations were observed for the high or intermediate exposure groups or for wheeze. In the RCM analysis, the estimated cumulative incidence of asthma was 16.1% in the background-exposed group and 26.7% in the very highly exposed group (Fisherian $p < 0.001$). Study limitations include reliance on an address-based categorical proxy for prenatal PFAS exposure, which likely results in non-differential exposure misclassification and limits the ability to distinguish prenatal from early-childhood exposure effects.

## Conclusions

In this study, very high prenatal PFAS exposure was associated with a higher incidence of childhood asthma. Although these results should be replicated, they suggest an important public health impact of AFFF-associated PFAS contamination.

---

## Author summary

### Why was this study done?

- Per- and polyfluoroalkyl substances (PFAS) are environmental contaminants that can affect the immune system and may contribute to the development of asthma.

- Previous epidemiological studies of PFAS and asthma have had inconclusive results and have mainly included populations exposed to low levels of PFAS.

- We investigated this question in Ronneby, a town in southern Sweden where some residents were exposed to very high PFAS levels via contaminated drinking water for over 30 years.

### What did the researchers do and find?

- We used Swedish national health and population registers to follow 11,488 children born in Blekinge County between 2006 and 2013, including children from Ronneby who were exposed to PFAS.

- We estimated prenatal exposure using mothers' residential address history and identified asthma diagnoses and prescriptions from medical records.

- Children with very high prenatal PFAS exposure had a higher incidence of asthma than children with low exposure, even after accounting for family and socioeconomic factors.

imputation by chained equations; PFAS, per- and polyfluoroalkyl substances; RCM, rubin causal model; RECORD, Reporting of Studies Conducted using Observational Routinely-Collected Data.

> **What do these findings mean?**
>
> - These findings suggest that very high exposure to PFAS is associated with a higher incidence of asthma in children.
>
> - Because effects were only seen at very high exposure levels, the results may not apply to populations exposed to lower PFAS levels.
>
> - Our exposure measure was based on residential addresses, and because many children lived at an exposed address after birth, we cannot fully distinguish between prenatal and early-life exposure.

## 1. Introduction

Asthma is a chronic airway inflammatory disorder characterized by variable expiratory airflow limitation and persistent respiratory symptoms, including wheeze, cough, shortness of breath, and chest tightness [1]. As the most common non-communicable disease in children, it poses a major global health challenge [2]. It is a substantial contributor to childhood hospitalizations, missed school days and missed work days for caregivers, and lower quality of life among both children and caregivers [3–5]. Asthma also places a large economic burden on national healthcare and welfare systems, due in part to its typical onset in childhood resulting in costs accrued over the life span [1,5,6].

The global prevalence of asthma has increased over the past 50 years and continues to rise in some areas of the world [7]. The reasons for this increase are not fully understood, but are thought to be driven in part by environmental exposures [7,8]. Early-life exposure to air pollution, passive smoking, and some microbial infections have been consistently linked to an increased asthma risk [4]. A possible role of environmental chemical exposures is less well-understood but plausible, given the potential immunotoxic effects of early-life chemical exposures [9].

Per- and polyfluoroalkyl substances (PFAS) are a class of several thousand fluorinated substances that have been widely used in industrial and consumer applications since their introduction in the early 1950s due to their desirable physical properties, including chemical and thermal stability and water and oil repellency [10]. However, growing awareness of the long elimination half-lives and extreme environmental persistence of PFAS has increased concern regarding their potential health effects [11]. We now know that children can be exposed to high levels of PFAS both prenatally and in early life, as PFAS cross the placental barrier and are also transferred into breastmilk [12,13]. These developmental exposures are particularly concerning because PFAS are endocrine-disrupting chemicals (EDCs) [14–16], and exposure to EDCs during sensitive development windows has been associated with numerous health risks [17]. While most individuals are exposed to background levels of PFAS via their diet, drinking water, and contact with consumer products, communities living near point sources of PFAS contamination often have elevated PFAS exposures [18].

A growing body of evidence has linked early-life PFAS exposure to immunosuppressive effects, including reduced antibody response to vaccination and an

increased risk of childhood infections [11,19–21]. Although a potential link between PFAS and hypersensitivity-related diseases like asthma is biologically plausible [20,22], results from epidemiological studies of PFAS exposure and asthma are inconclusive [23]. In its most recent PFAS risk assessment, the European Food Safety Agency concluded that epidemiological studies provided insufficient evidence to conclude on associations between PFAS and asthma [23]. However, previous studies have been limited by sample size and have only been conducted at levels of PFAS exposure found in the general population. There are no studies of potential effects of PFAS in highly-exposed populations.

To address this gap, we conducted a register-based study of the effects of prenatal PFAS exposures on the incidence of clinically-diagnosed asthma in children born in Blekinge county, Sweden. A subset of residents in the city of Ronneby, located in Blekinge county, were highly exposed to PFAS for over 30 years from aqueous film-forming foam (AFFF) contamination in their drinking water. We hypothesized that children born to mothers who lived in the highly-exposed area of Ronneby would have increased incidence of wheeze and asthma.

## 2. Materials and methods

### 2.1. Setting

In December 2013, it was discovered that one of two municipal waterworks in Ronneby, Sweden was contaminated by high PFAS due to AFFF runoff from a local military airfield. The total PFAS concentration in the outgoing drinking water was 10,380 ng/L, compared to 48 ng/L in the second Ronneby waterworks and less than 5 ng/L in the water provided in a neighboring municipality [24]. The highly contaminated waterwork was immediately shut down and all water in Ronneby was switched to the second municipal waterwork. However, by this point, approximately ⅓ of Ronneby households had unknowingly consumed highly contaminated water for decades. Biomonitoring of 3,293 Ronneby residents in 2014 and 2015 found extremely elevated PFAS serum levels, even compared to other AFFF-exposed communities [24].

### 2.2. Study cohort

The study cohort is a register-based open cohort of all children born between 2006 and 2013 in Blekinge County, Sweden, which includes Ronneby municipality. Records across several Swedish population and health registries were linked using personal identity numbers [25]. The study population was identified from the Total Population Register, which includes information on residential address ascertained on December 31 of each calendar year and includes over 99.9% of children born in Sweden [25]. Links between parents and children were ascertained from the Swedish Multi-Generation Register [26]. Additional covariate information was linked from the National Medical Birth Register and the Longitudinal Integrated Database for Health Insurance and Labor Market Studies, which is reported annually [27,28].

Clinical diagnosis data was extracted from the National Patient Register, which contains medical information for all in- and outpatient specialist visits including ICD-10 codes and admission dates. All national in-patient visits have been recorded since 1987 and specialized outpatient care was added to the National Patient Register in 2001 [29,30]. Prescription data was extracted from the National Prescribed Drug Register, which records information on all pharmacy-dispensed medications, including anatomical therapeutic chemical codes and dispensation dates. Reporting to the National Prescribed Drug Register began in July 2005, but consistent reporting was not achieved until 2006 [31]. Primary healthcare records, which were not consistently reported electronically until 2010, were obtained from the Blekinge Healthcare Register and used to validate the outcome algorithms.

The final study cohort included children born in Blekinge county between 2006 (the first year with records available in the National Prescribed Drug Register) and 2013 (the last year of high PFAS exposure). All children were followed from birth until outcome incidence, death (N = 24), or censoring. Children were censored at emigration from Sweden, the outcome-specific maximum age, or the end of the study on December 31, 2022. A summary of the primary study cohort and validation cohorts and their data sources is provided in S1 Table.

Data linkage was performed using personal identity numbers and then de-identified by Statistics Sweden prior to delivery to the research team. The Swedish Ethical Review Authority approved the study (2021–04872) and waived the requirement for individual informed consent because the research used existing registry data and could not practicably be conducted if consent were required. The study was conducted within a broader ethically approved research program; however, no study-specific prospective protocol or statistical analysis plan was developed for the present investigation.

## 2.3. Exposure assessment

Prenatal exposure to PFAS was estimated using a proxy variable based on maternal address during the five calendar years preceding the year of delivery. Maternal residential address data from the Total Population Register were linked to annual municipal water distribution records showing which addresses had been provided with water from the highly contaminated waterworks. Prenatal exposure was categorized as "very high" if the child's mother lived at an address in Ronneby supplied with highly contaminated water for all five years preceding the year of delivery; "high" if she lived at an address in Ronneby with contaminated water for at least one of those five years; "intermediate" if she lived in Ronneby during that period but never at an address supplied with highly contaminated water; and "background" if she did not live in Ronneby during the five years preceding the year of delivery (Table 2). As a proxy for fetal PFAS exposure, this categorization assumes that maternal residential history reflects drinking water exposure before and during pregnancy, and does not account for individual differences in water consumption or temporal changes in PFAS concentrations. Although earlier Ronneby registry studies used a three-level exposure categorization [32,33], we applied a four-level categorization supported by exposure validation analyses indicating stepwise increases in measured PFAS concentrations across the four groups.

We validated the exposure categories using a subset of the Ronneby Biomarker Cohort. This cohort, described in detail in Xu and colleagues (2021), measured PFAS concentrations in serum samples collected between 2014 and 2016 from 3,523 participants of all ages from Ronneby and a reference group [24]. We limited our validation cohort to women with a known residential history the five years before sampling who were a similar age to mothers in our primary study cohort (21–40 years) (S1 Table). We assigned a categorical exposure level to each participant in the validation cohort using the method described above. We then compared measured serum PFAS concentrations across the four different categorical exposure groups (Table 2). Written informed consent was received from all participants in the Ronneby Biomarker Cohort, which was approved by the Regional Ethical Review Board in Lund, Sweden (number 2014/4).

## 2.4. Outcome assessment

Swedish healthcare is publicly funded and free of charge for all children up to the age of 18 years [34]. Three asthma-related outcomes were ascertained based on ICD-10 diagnosis codes from the National Patient Register and prescription drug dispensations in the National Prescribed Drug Register. We estimated asthma incidence through age 12 using an algorithm that was previously validated in the Swedish pediatric population [35], where the date of incidence was considered to be either the first occurrence of an asthma diagnosis in the National Patient Register (ICD-10 code J45) or the first dispensation of asthma medication, with a requirement for at least one repeated dispensation.

To address the challenge of diagnosing asthma in early childhood and account for the Swedish Pediatric Society's separate asthma diagnosis guidelines for children under and over 36 months of age [36,37], we created a second, stricter asthma algorithm that additionally required at least one asthma diagnosis or dispensation of asthma-related prescription drugs after age 36 months (3+ years). Incidence of this strict asthma outcome was also considered to be the first occurrence of asthma diagnosis or dispensation of asthma medication, and was ascertained through age 12.

Finally, previous studies have found associations between PFAS and early-life wheeze [8,38,39], which may progress into asthma in some but not all cases [37]. Because asthma and wheeze can be difficult to distinguish in children under three, we created a broader outcome to capture early-life wheeze between ages 0–36 months. This outcome included

diagnosis codes for asthma (J45) and acute lower respiratory infections (J20-22) and/or at least one dispensation of asthma- and wheeze-related medication, to encompass both early asthma presentations and transient wheezing illnesses.

We refer to these three outcomes as "asthma", "asthma (3+)", and "wheeze," respectively. Their specific algorithms are described in detail in the Supplemental Material (S2 Table).

We validated our three outcome algorithms using primary care records from the Blekinge Healthcare Register. Early-life asthma is typically diagnosed in primary care [37], making diagnosis codes from primary care records an appropriate gold standard for algorithm validation. To validate our outcome algorithms in a large cohort, we created a prospective birth cohort of children born in Blekinge between 2010 and 2021 and followed them until outcome incidence, death, or until censoring by moving out of Blekinge county, the outcome-specific maximum age, or the end of study on December 31, 2022 (S1 Table). For each child, we identified outcome incidence using both our primary outcome algorithms and ICD-10 codes from primary care records. We then calculated outcome-specific specificity and sensitivity. The algorithms used to estimate outcomes from primary care records are included in S2 Table.

## 2.5. Covariate assessment

Although exposure to PFAS-contaminated water in Ronneby was not influenced by individual characteristics beyond residential address, we observed differences in personal characteristics across exposure groups due to underlying neighborhood-level differences. Therefore, we identified potential confounders of our hypothesized association between PFAS exposure status and the incidence of asthma and/or wheeze in childhood using a directed acyclic graph, DAG (S1 Fig). Maternal smoking status in early pregnancy, parity, and age at delivery were obtained from the Medical Birth Register [40–42]. Family disposable income and the highest-achieved maternal education at the year of delivery were obtained from the Longitudinal Integrated Database for Health Insurance and Labor Market Studies [43,44]. An indicator variable for having at least one foreign-born parent was obtained from the Total Population Register, and was included because this may impact healthcare-seeking behavior [45,46], and it varied by PFAS exposure group due to neighborhood-level differences in the number of foreign-born residents. Parental asthma status (yes: the child's mother and/or father had asthma; no: neither parent had asthma) is an important predictor of child asthma [47] and may also be associated with socioeconomic status and the neighborhood of residence. Therefore, we determined the asthma status of each child's mother and father using the same algorithm that we used to identify incident cases in children in the main analysis [35]. A parent was considered to have asthma (yes or no) if they met the diagnosis criteria for asthma at any point in the study (2006–2022), and the final parental asthma status was considered as an indicator variable where "yes" indicated that at least one parent had asthma, and "no" indicated that neither parent had asthma.

## 2.6. Statistical analysis

Our exposure validation analysis used Jonckheere–Terpstra tests to assess trends in PFAS concentrations across the ordered categorical exposure groups [48,49]. In the outcome validation analysis, we calculated outcome-specific sensitivity and specificity where the "true" diagnosis was ascertained from the Blekinge primary care records and the "test" diagnosis was determined from our algorithms using the National Patient Register and National Prescribed Drug Register records.

We used the Kaplan–Meier method to visualize the cumulative incidence of each outcome by prenatal PFAS exposure group, without adjustment for confounding variables. For formal inference, we applied Cox proportional hazard models to estimate the association between prenatal PFAS exposure and disease incidence, using robust variance estimators to account for potential residual correlation within siblings [50,51]. All models used age as the time axis and stratified the baseline hazard by sex and parity (primiparous or multiparous) to account for non-proportional hazards. Adjusted models included the following covariates: maternal smoking status in early pregnancy (smoker or non-smoker); maternal education at the year of delivery (primary and lower secondary, upper secondary, and post-secondary); an indicator variable

for at least one foreign-born parent (yes or no) and parental asthma (yes or no); family disposable income at the year of delivery (quartiles); and maternal age at delivery (quartiles). Primary models were limited to children with complete covariate information. The proportional hazards assumption was evaluated for each covariate using scaled Schoenfeld residuals regressed against a Kaplan–Meier transformation of time [52]. We formally assessed the proportional hazards assumption for our categorical exposure variable by jointly testing the time-by-exposure interaction terms using robust Wald tests, which account for the cluster-robust covariance estimates from the fitted Cox models; this assessment was added in response to peer review.

To assess the impact of missing data on our estimated associations, we used multiple imputation by chained equations (MICE) to impute missing covariates. For each outcome, we generated 20 imputed datasets with 20 iterations per dataset. The imputation model included all covariates used in our Cox proportional hazards models, additional potential predictors of missing covariates, the outcome-specific event indicator, and the Nelson–Aalen estimate of the marginal cumulative hazard evaluated at each individual's observed follow-up time [53–55]. We then fit Cox proportional hazards models with cluster-robust standard errors for each imputed dataset and pooled the estimates using Rubin's rules [53,56].

PFAS are known EDCs with sex-specific effects [57], and therefore we hypothesized that sex may modify any association between PFAS and asthma. To test this, we evaluated an interaction term between prenatal PFAS exposure and child sex using a robust Wald test for the sex-interaction terms for each outcome. We then repeated all analyses stratified by child sex to compare the sex-specific estimated hazards.

Following the primary survival analyses, we conducted a secondary analysis using a Rubin Causal Model (RCM) potential outcomes framework to assess the possible causal effect of very high versus background prenatal PFAS exposure on childhood wheeze and asthma. We constructed a hypothetical randomized experiment by matching each child in the very high exposure group to three controls from the background group using 3:1 nearest neighbor matching with robust rank-based Mahalanobis distance [58–61]. Matching included the same confounders as our primary survival models and was restricted to children with complete covariate data. We estimated the difference in cumulative incidence by the end of follow-up between the two matched exposure groups with Kaplan–Meier survival models. We then tested the Fisher sharp null hypothesis which states that each child's outcome would have been identical under both exposure levels (formally $H_0 : Y_i(W_i = 0) = Y_i(W_i = 1)$, for all children (i) using the difference in cumulative incidence as our test statistic [62,63]. Under this null hypothesis, treatment assignment can be viewed as hypothetically randomized within each matched set of four children, allowing us to approximate one-sided Fisher's exact p-values for each outcome with 100,000 exposure permutations [64,65]. Causal interpretation of this secondary analysis relies on the assumptions outlined in Imbens and Rubin (2015): the stable unit treatment assignment value assumption, where there is no interference between units and the treatment is well-defined (i.e., consistency); probabilistic assignment mechanism, where every unit has a non-zero probability of each treatment; and unconfoundedness, where exposure is independent of potential outcomes after matching on background covariates [63].

All analyses were performed in R version 4.5.1 (2025-06-13) using the following packages: "Tidyverse" version 2.0.0 [66], "survival" version 3.8.3 [67], "survminer" version 0.5.1 [68], "car" version 3.1.3 [69], "mice" version 3.18.0 [53], and "MatchIt" version 4.7.2 [70]. We reported our study following the Reporting of Studies Conducted using Observational Routinely-Collected Data (RECORD) Statement (S1 RECORD Checklist) [71].

## 3. Results

Of the 12,585 children in our cohort born in Blekinge between 2006 and 2013, 11,488 (91%) had complete covariate information and were included in our final study population. The variable with the highest rate of missingness was maternal smoking status during early pregnancy (6%). Individuals with missing covariate information were more likely to have a lower family disposable income, at least one parent born abroad, and to be categorized in the background exposure group (S3 Table).

Baseline characteristics of the final study population are shown in Table 1. Most children in the study had older siblings (58%), were born to non-smoking mothers (92%), and had parents who were both born in Sweden (81%). The median maternal age at delivery was 30 years (interquartile range, IQR: 26–34). Overall, 17% of children in the study had at least one parent with prevalent asthma. Nearly one-quarter (24%) of children had at least one maternal sibling also included in the study population.

The study included 194 children (2%) in the very high prenatal exposure group, 479 children (4%) in the high prenatal exposure group, 1,591 children (14%) in the intermediate exposure group, and 9,224 children (80%) in the background exposure group. Children in the very high exposure group were less likely to have at least one parent born abroad (9% versus 19% overall). Their mothers were more likely to have only completed primary or lower-secondary education (32% versus 20% overall) and to have had a previous pregnancy (75% versus 58% overall) (Table 1).

In the exposure validation cohort ($N = 209$), PFAS concentrations increased across exposure categories for the three PFAS compounds measured in the cohort (perfluorooctane sulfonic acid, PFOS; perfluorohexane sulfonic acid, PFHxS; and perfluorooctanoic acid, PFOA; Table 2 and S2 Fig), and Jonckheere-Terpstra tests indicated a strong trend for all three PFAS ($p < 0.001$). For example, the median PFHxS concentration in the very high exposure group was 164.8 ng/mL (IQR: 106.1, 283.0) compared to 100.0 ng/mL (IQR: 64.8, 180.3) in the high exposure group, 30.5 ng/mL (IQR: 21.3, 102.4) in the intermediate group, and 0.8 ng/mL (IQR: 0.7, 1.1) in the background group.

A total of 2,653 study children (23%) had at least one outcome. Wheeze had the highest overall prevalence in the study population at 18%, while the prevalence of asthma was 17%, and the prevalence of asthma (3+) was 13%. For children with at least one outcome, it was most common to have all three outcomes ($N = 891$; 34%) or to just have wheeze ($N = 657$; 25%) (S3 Fig). In the outcome validation cohort of 16,145 children born in Blekinge between 2010 and 2021, our outcome algorithms based on the National Patient Register and National Prescribed Drug Register performed well compared to primary care records, with outcome-specific sensitivities between 0.76 and 0.86 and specificities between 0.93 and 0.97 (S4 Table).

Unadjusted cumulative incidence curves indicated higher cumulative incidence of asthma and asthma (3+), but not wheeze, in children with very high prenatal exposure (Fig 1). Similar results were also found in our unadjusted Cox proportional hazard models. In our fully adjusted models, we observed an increased hazard in the very high prenatal exposure group compared to background for asthma (HR: 1.44, 95% CI [1.08, 1.92]) and asthma (3+) (HR: 1.59, 95%

Table 1. Cohort baseline characteristics, displayed as *N* (%) or median [interquartile range].

| Variable | Overall | Prenatal exposure group | | | |
|---|---|---|---|---|---|
| | | Very High | High | Intermediate | Background |
| N | 11,488 | 194 | 479 | 1,591 | 9,224 |
| Maternal smoking in early pregnancy (Yes) | 901 (7.8) | 19 (9.8) | 82 (17.1) | 132 (8.3) | 668 (7.2) |
| Parity (Multiparous) | 6,674 (58.1) | 146 (75.3) | 259 (54.1) | 908 (57.1) | 5,361 (58.1) |
| Sex (Female) | 5,539 (48.2) | 87 (44.8) | 238 (49.7) | 799 (50.2) | 4,415 (47.9) |
| Maternal education | | | | | |
| Primary and lower secondary | 2,292 (20.0) | 62 (32.0) | 135 (28.2) | 307 (19.3) | 1,788 (19.4) |
| Upper secondary | 3,630 (31.6) | 79 (40.7) | 175 (36.5) | 508 (31.9) | 2,868 (31.1) |
| Post secondary | 5,566 (48.5) | 53 (27.3) | 169 (35.3) | 776 (48.8) | 4,568 (49.5) |
| Maternal age at delivery | 30.0 [26.4, 33.7] | 30.9 [26.8, 34.9] | 27.7 [24.4, 32.0] | 30.1 [26.7, 33.7] | 30.1 [26.5, 33.7] |
| At least one parent born abroad (Yes) | 2,223 (19.4) | 18 (9.3) | 75 (15.7) | 273 (17.2) | 1857 (20.1) |
| Annual family disposable income (Swedish krona × $10^5$) | 3.8 [3.0, 4.5] | 3.7 [3.0, 4.4] | 3.5 [2.7, 4.1] | 3.8 [2.8, 4.4] | 3.8 [3.0, 4.6] |
| Parental asthma (Yes) | 1963 (17.1) | 37 (19.1) | 106 (22.1) | 250 (15.7) | 1,570 (17.0) |

**Table 2. Validation of the prenatal exposure category definitions using measured PFAS concentrations (ng/mL) from a subset of the Ronneby Biomarker Cohort (female participants between ages 21 and 40 years, *N*=209) [24].**

| Exposure category | Definition | N | PFAS, ng/mL: median [interquartile range] | | |
|---|---|---|---|---|---|
| | | | PFOS | PFHxS | PFOA |
| Very high | Registered at an address receiving contaminated water for all five years preceding the year of delivery. | 80 | 218.7 [140.4, 344.1] | 164.8 [106.1, 283.0] | 11.8 [7.2, 18.9] |
| High | Registered at an address receiving contaminated water for at least one year in the five years preceding the year of delivery, but does not meet criteria for very high exposure. | 54 | 137.4 [80.7, 194.8] | 100.0 [64.8, 180.3] | 7.3 [3.7, 14.2] |
| Intermediate | Registered at an address in Ronneby for at least one year in the five years preceding the year of delivery, but does not meet criteria for very high or high exposure. | 39 | 47.7 [30.8, 121.1] | 30.5 [21.3, 102.4] | 3.8 [2.0, 5.6] |
| Background | Registered at an address in Blekinge County, but never lived in Ronneby. | 36 | 3.6 [2.6, 4.8] | 0.8 [0.7, 1.1] | 1.6 [0.8, 1.9] |

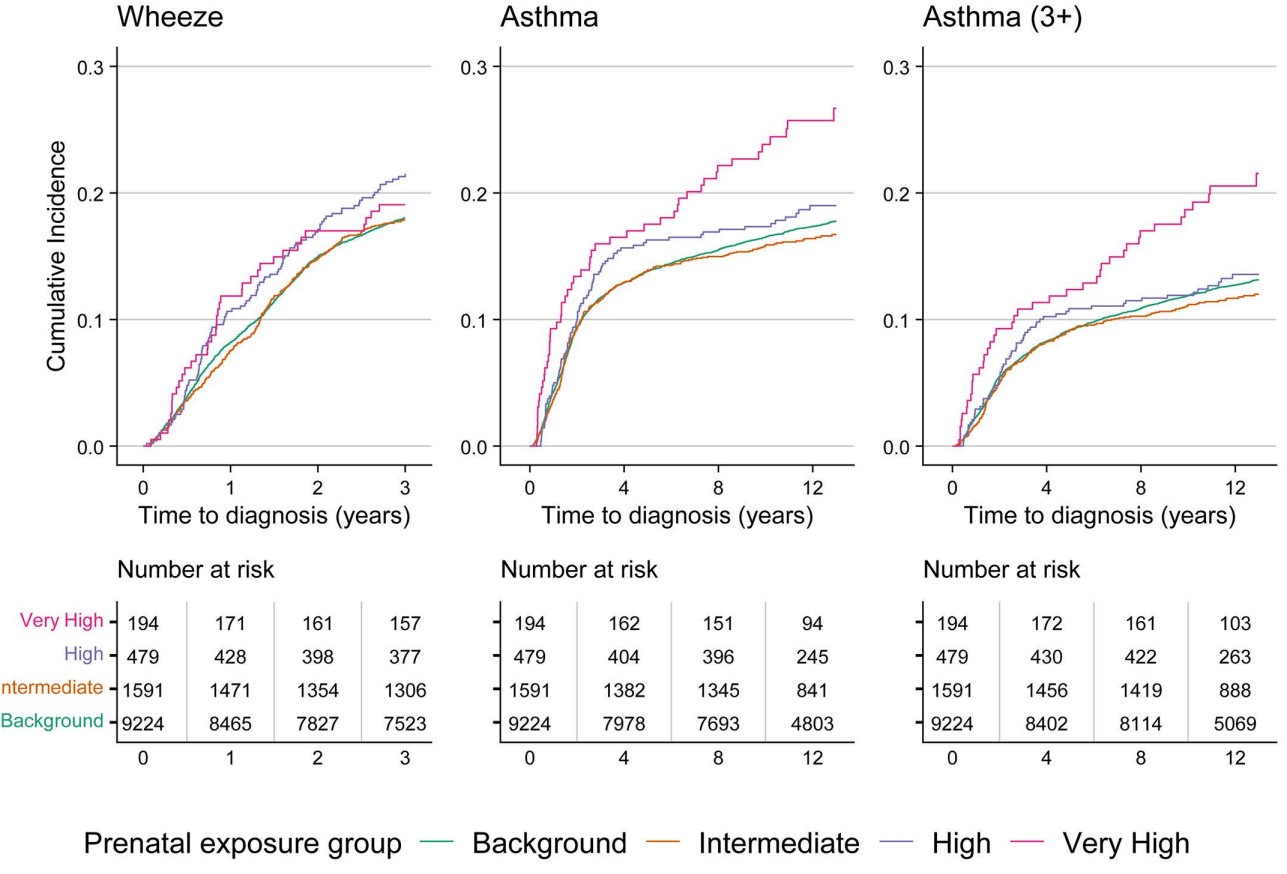

**Fig 1. Cumulative incidence by outcome, estimated using the Kaplan–Meier method.**

CI [1.14, 2.21]) but not in the high and intermediate exposure groups (Table 3). Other model covariates, including parental asthma, maternal smoking status in early pregnancy, and having at least one parent born abroad, were also associated with our outcomes (S4 Fig). For all outcomes, robust Wald tests for the exposure-by-time interaction terms indicated no evidence of non-proportional hazards by exposure category (wheeze: $p = 0.71$; asthma: $p = 0.14$; asthma (3+): $p = 0.29$).

Pooled hazard ratios (HRs) estimated from the multiple-imputed datasets were similar to the primary model results (S5 Fig and S5 Table). In models including sex-by-exposure interaction terms, the interaction $p$-value was lower for wheeze ($p = 0.08$) than for asthma ($p = 0.33$) or asthma (3+) ($p = 0.52$). In sex-stratified models for wheeze, the HR comparing very high with background exposure differed in direction between girls (HR: 0.50, 95% CI [0.24, 1.04]) and boys (HR: 1.23, 95% CI [0.84, 1.79]) (S6 Table and S6 Fig).

In our secondary analysis, our matched subcohort included all 194 very high-exposed children with complete covariates (out of 202 total high-exposed children) and 582 matched background-exposed children. The subcohort had excellent covariate balance between exposed and background individuals, with a standardized mean difference of 0.02 or less for all covariates (S7 Fig). We found a higher cumulative incidence of asthma and asthma (3+) by age 12 in the very high-exposed group compared to the matched background-exposed group (Fig 2). The estimated cumulative incidence of asthma was 10.6 percentage points higher in the very high exposure group compared to background (26.7% versus 16.1%; Fisherian $p < 0.001$) and the estimated cumulative incidence of asthma (3+) was 10.4 percentage points higher in the very high exposure group compared to background (21.5% versus 11.1%; Fisherian $p < 0.001$). The cumulative incidence of wheeze was comparable between the two exposure groups and we could not reject the null hypothesis (Fisherian $p = 0.6$). The null randomization distributions of the difference in cumulative incidence for each outcome are shown in S8 Fig.

**Table 3. Hazard ratios of disease incidence by prenatal exposure group.**

| Prenatal exposure group | Events | Person-years | Hazard ratio (95% confidence interval) | |
|---|---|---|---|---|
| | | | Simple model[1] | Adjusted model[2] |
| **Wheeze** | | | | |
| Background | 1,661 (18%) | 24,678 | – | – |
| Intermediate | 284 (18%) | 4,269 | 1.00 (0.87, 1.14) | 0.99 (0.87, 1.14) |
| High | 103 (22%) | 1,255 | 1.22 (0.98, 1.52) | 1.08 (0.87, 1.35) |
| Very high | 37 (19%) | 509 | 1.08 (0.76, 1.53) | 0.97 (0.69, 1.37) |
| **Asthma** | | | | |
| Background | 1,596 (17%) | 95,960 | – | – |
| Intermediate | 261 (16%) | 16,728 | 0.95 (0.83, 1.09) | 0.95 (0.83, 1.09) |
| High | 89 (19%) | 4,925 | 1.09 (0.86, 1.39) | 0.96 (0.75, 1.22) |
| Very high | 50 (26%) | 1,919 | 1.56 (1.16, 2.08) | 1.44 (1.08, 1.92) |
| **Asthma (3+)** | | | | |
| Background | 1,170 (13%) | 100,518 | – | – |
| Intermediate | 186 (12%) | 17,531 | 0.92 (0.79, 1.08) | 0.92 (0.79, 1.08) |
| High | 63 (13%) | 5,207 | 1.05 (0.80, 1.37) | 0.94 (0.72, 1.23) |
| Very high | 40 (21%) | 2,033 | 1.68 (1.21, 2.32) | 1.59 (1.14, 2.21) |

[1]Baseline hazard is stratified by child sex.

[2]Baseline hazard is stratified by child sex and parity (primiparous or multiparous). The model is also adjusted for the following covariates: maternal smoking status in early pregnancy (smoker or non-smoker); maternal education (primary and lower secondary, upper secondary, and post-secondary); at least one foreign-born parent (yes or no); family disposable income (quantiles); maternal age at delivery (quantiles), and maternal asthma (yes or no).

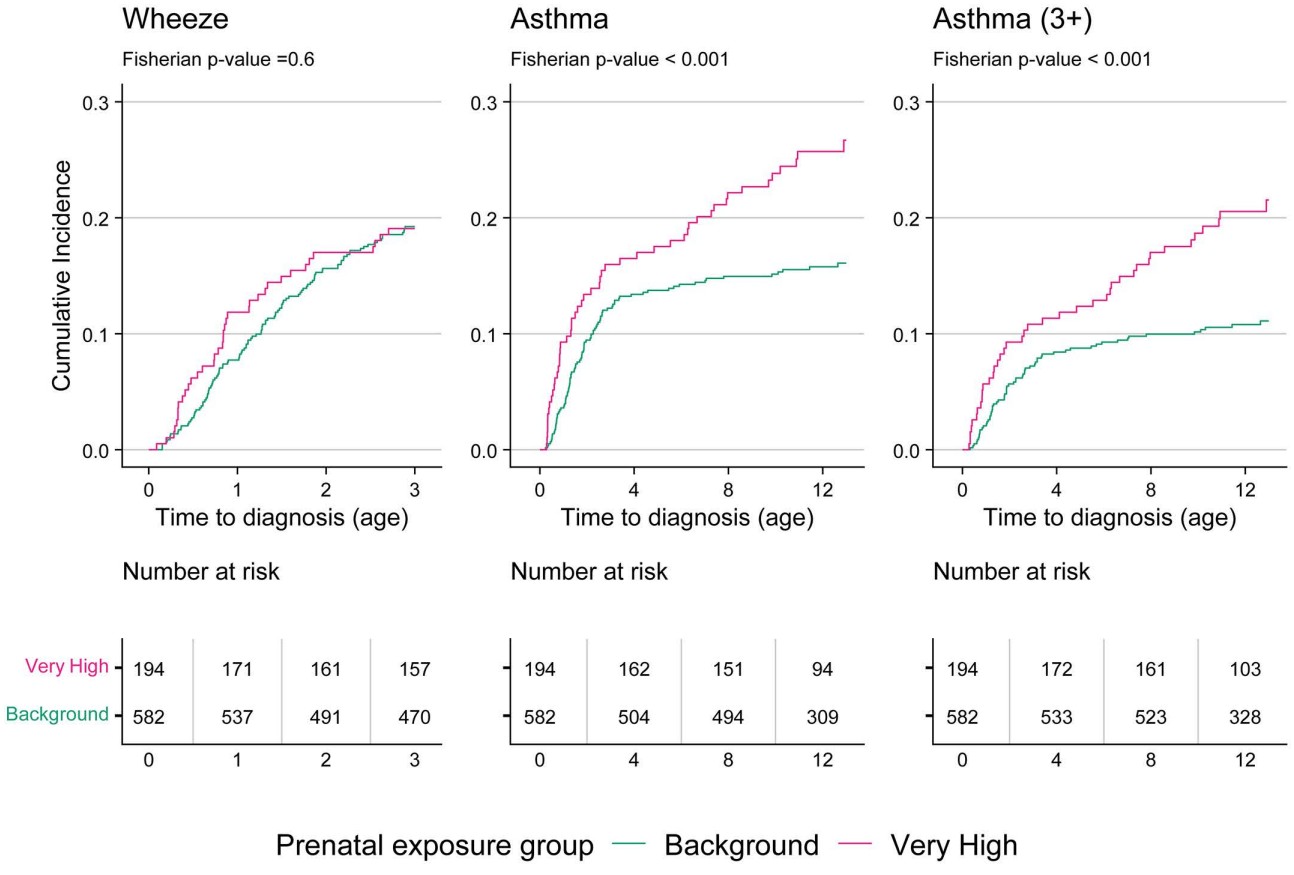

**Fig 2. Cumulative incidence by outcome in the matched subcohort (*N*=776), estimated using the Kaplan–Miller method.**

## 4. Discussion

In this large registry-based cohort of children born in Blekinge County, Sweden, very high prenatal PFAS exposure was associated with higher incidence of asthma and asthma (3+) when compared to background levels of prenatal PFAS exposure, but we found no apparent effect of very high PFAS exposure on early-childhood wheeze. We also could not reject the null hypothesis when examining associations between intermediate and high PFAS exposure compared to background exposure for any of the outcomes after covariate adjustment. In our secondary RCM analysis, we similarly found higher cumulative incidence of asthma and asthma (3+) in the highly-exposed population compared to a matched control population of background-exposed individuals.

An association between high prenatal PFAS exposure and increased incidence of asthma is biologically possible, as lung development, which begins early in embryonic development and extends through adolescence, is highly sensitive to disruption by environmental toxicants because of its dependence on coordinated developmental processes [22]. PFAS concentrations in both mice and human fetuses are highest in lung tissue [72,73], suggesting that the lungs may be an important target of prenatal PFAS toxicity. In vitro studies have found that PFAS exposure aggravates mast cell-mediated allergic inflammation [74–77] and at high concentrations, PFAS may exacerbate airway hypersensitivity reactions [20]. In vivo studies have demonstrated that high- to moderate-PFAS exposures can aggravate allergic lung responses in ovalbumin-sensitized mice and shift cytokine production towards a T-helper cell (T$_H$2)-dominated response [77–80]. Prenatal PFOS exposure in rats has been shown to inhibit perinatal lung development [81], induce oxidative injury and

apoptosis leading to histopathological changes in the lung [82], reduce alveolar numbers and increase lung inflammation [83], and change gene expression in the lung [84].

Despite this experimental evidence, results from epidemiological studies are mixed. Cross-sectional studies of children and adolescents have generally identified a positive association between serum or plasma PFAS concentrations and self-reported current or past asthma [85–91], although other cross-sectional studies failed to reject the null hypothesis [39,57,92,93]. Longitudinal studies of prenatal PFAS exposure and the prevalence of asthma and/or wheeze in childhood have had mixed results, with most studies identifying null or weakly negative associations [39,94–100] while others have identified positive associations [95,101,102]. However, previous studies have varied considerably on the specific outcome definitions used and the timing of outcome ascertainment, making it difficult to directly compare results. Almost all previous longitudinal studies have relied on parental-reported outcomes, and all studies except one [103] have included less than 3,000 participants.

Our study differs from previous epidemiological investigations by including a subset of participants with very high exposure to AFFF-associated PFAS, yielding a much wider exposure range than typically observed in population-based studies. This community-wide contamination from drinking water allows evaluation of potential health effects at PFAS concentrations relevant to highly exposed populations worldwide. Although PFAS concentrations could not be measured directly in the study participants, we validated our exposure classification in a separate cohort of women of childbearing age. Samples in the validation cohort were collected after the contaminated drinking water supply had been identified and replaced (2014–2016), which likely led to some underestimation of true exposure levels in our study population. Even with this potential underestimation, concentrations of AFFF-related PFAS in the very-high exposure group were several hundred times higher than in other epidemiological birth cohorts. For example, the median PFHxS concentration in our very-high exposure group (164.8 ng/mL) far exceeded those reported in the Japan Environment and Children's study (0.34 ng/mL), in the Danish Odense Child cohort (0.36 ng/mL), and in the American Healthy Heart Study (0.7 ng/mL) [95,102,103]. The fact that associations with asthma were detected only among children with very high prenatal PFAS exposure may indicate a threshold effect that was not observable in background-exposed populations.

Our study has several limitations. Although we used detailed Swedish registries to accurately adjust for important confounding factors like maternal age at delivery and maternal smoking in pregnancy [104], we cannot rule out other sources of unmeasured confounding like parental smoking status after delivery. Our study could not account for parental history of childhood asthma, but we adjusted for prevalent parental asthma during the study period. Other sources of environmental contamination, including air pollution and noise, are similar across all study participants and are unlikely to confound the observed PFAS-asthma association.

Our study also has some limitations in outcome ascertainment. Asthma in early childhood can be difficult to diagnose because inflammation and small airway obstruction can cause symptoms that overlap with transient wheezing from common viral infections. Diagnostic accuracy improves after age three, when transient causes become less common. We addressed this by defining three outcomes: wheeze, asthma, and asthma (3+). In our registry data, the wheeze outcome included medical records with asthma (ICD-10 J45) and acute lower respiratory infections (J20-J22), making this outcome less specific than the asthma and asthma (3+) outcomes. This may have made it more difficult to detect any true PFAS-wheeze association.

A final study limitation is its address-based categorical exposure assessment. This approach may induce non-differential exposure error with both a classical-like error structure (i.e., inability to account for temporal changes in PFAS concentrations in drinking water) and Berkson-like error structure (i.e., inability to account for individual differences in contaminated water consumption) [105]. These errors are generally expected to increase the variance of effect estimates and bias them towards the null [106]. Furthermore, the exposure assessment is based on maternal exposure prior to delivery and does not capture ongoing exposure in childhood. Many children with high or very high prenatal exposure continued living in the contaminated area through 2013; for example, at age three, more than half (57%) of the children in the very

high prenatal exposure group still resided at an address receiving contaminated water. Given the likelihood of continued postnatal exposure, we cannot attribute the observed associations solely to prenatal exposure, and further research is needed to clarify which developmental periods are most sensitive to high PFAS exposure.

Despite these limitations, our address-based exposure assessment also offers several advantages. Proxy measures based on residential history are less prone to confounding from individual-level factors that influence both personal exposure and disease risk [105], and the use of categorical exposure groups rather than continuous individual-level exposure estimates may shift exposure error from a classical-like to a Berkson-like structure, thereby reducing bias in the estimated exposure-response association [107].

This study has several additional strengths. The large population-based cohort ($N = 11,488$) included detailed longitudinal information on time-to-event and censoring, permitting survival analysis of asthma incidence and efficient use of follow-up time [108,109]. Unlike many previous studies that relied on parental-reported outcomes subject to recall bias, our study used clinically relevant outcomes ascertained from detailed medical and drug dispensation records with high validity. Our study included all children born in Blekinge county and therefore is unlikely to be impacted by selection bias. In addition, healthcare is free for all children in Sweden and is easily accessible across Blekinge county, reducing the likelihood of differential outcome detection. Finally, linkage to multiple national registries provided extensive individual-level data, enabling adjustment for key confounders.

Overall, our findings indicate that children with very high prenatal PFAS exposure from AFFF-contaminated drinking water experienced greater asthma incidence than those with background exposure. These findings likely have limited generalizability to populations with only background-level exposures or different PFAS mixtures, as exposure-response relationships remain uncertain and individual compounds vary in toxicity. However, AFFF-related PFAS contamination is a major source of high environmental exposure globally [18,110], and evidence from Ronneby offers important insights into the potential health effects of such contamination in affected communities. Replication in other highly exposed populations is needed to confirm these results, but they point to a substantial and previously unrecognized public health consequence of AFFF-related PFAS contamination.

## Supporting information

**S1 RECORD Checklist. The REporting of studies Conducted using Observational Routinely-collected health Data (RECORD) checklist.**
(DOCX)

**S1 Table. Summary of the different cohorts used in the study.**
(DOCX)

**S2 Table. Outcome definitions.**
(DOCX)

**S3 Table. Baseline characteristics for children with and without complete covariate data.**
(DOCX)

**S4 Table. Outcome algorithm performance compared to primary care records.**
(DOCX)

**S5 Table. Pooled hazard ratios from the 20 multiple-imputed datasets.**
(DOCX)

**S6 Table. Hazard ratios for each outcome stratified by sex.**
(DOCX)

**S1 Fig. Directed acyclic graph (DAG) for the association between PFAS exposure via the municipal water source and diagnosis of childhood asthma.**
(DOCX)

**S2 Fig. PFAS concentrations (ng/mL) in a subset of the Ronneby Biomarker Cohort.**
(DOCX)

**S3 Fig. UpSet plot of outcome combinations among children with at least one outcome ($N = 2,653$).**
(DOCX)

**S4 Fig. Hazard ratios for covariates from the primary adjusted models.**
(DOCX)

**S5 Fig. Hazard ratios from the primary complete case analysis ("Primary") compared to the multiple-imputed datasets ("MICE").**
(DOCX)

**S6 Fig. Hazard ratios in the primary models stratified by sex.**
(DOCX)

**S7 Fig. Love plot for the matched balanced cohort.**
(DOCX)

**S8 Fig. Null randomization distributions of the test statistic used for the approximation of the Fisher exact *p*-value.**
(DOCX)

## Acknowledgments

Views and opinions expressed are those of the authors only and do not necessarily reflect those of the European Union or the European Research Executive Agency (REA), and neither the European Union or the REA can be held responsible for them.

## Author contributions

**Conceptualization:** Annelise J. Blomberg, Anna Saxne Jöud.

**Data curation:** Annelise J. Blomberg.

**Formal analysis:** Annelise J. Blomberg.

**Funding acquisition:** Annelise J. Blomberg, Anna Saxne Jöud.

**Investigation:** Annelise J. Blomberg.

**Methodology:** Annelise J. Blomberg, Christel Nielsen, Beata Borgström Bolmsjö, Marie-Abèle Bind, Linda Hartman.

**Project administration:** Annelise J. Blomberg.

**Software:** Annelise J. Blomberg, Marie-Abèle Bind.

**Supervision:** Christel Nielsen, Linda Hartman, Anna Saxne Jöud.

**Validation:** Annelise J. Blomberg.

**Visualization:** Annelise J. Blomberg.

**Writing – original draft:** Annelise J. Blomberg.

**Writing – review & editing:** Annelise J. Blomberg, Christel Nielsen, Beata Borgström Bolmsjö, Marie-Abèle Bind, Linda Hartman, Anna Saxne Jöud.

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
