## [Editor Report · Decision Letter 0]

22 May 2025

Dear Dr Blomberg,

Thank you for submitting your manuscript entitled "Prenatal exposure to perfluoroalkyl substance (PFAS) and incidence of asthma and wheeze in childhood: A cohort study in Ronneby, Sweden" for consideration by PLOS Medicine.

Your manuscript has now been evaluated by the PLOS Medicine editorial staff and I am writing to let you know that we would like to send your submission out for external peer review.

For clinical studies, please upload a copy of your trial study protocol as a supporting information file. The study protocol should be the version submitted for approval to the institutional review board or ethics committee, should include any amendments to the study protocol, as well as the date of their approval by the institutional review or ethics committee. Please also detail any deviations from the study protocol in the Methods section of your manuscript. The editors will consider the protocol and study conduct prior to a final decision for external review.

Please re-submit your manuscript within two working days, i.e. by May 26 2025 11:59PM.

Kind regards,

Andreia Cunha, PhD

Senior Editor

PLOS Medicine

---

## [Decision Letter · Decision Letter 1]

15 Sep 2025

Dear Dr Blomberg,

Sincere apologies for the delay in getting back to you with a decision, which was due to challenges in securing the necessary Reviewers. Many thanks for submitting your manuscript "Prenatal exposure to perfluoroalkyl substance (PFAS) and incidence of asthma and wheeze in childhood: A cohort study in Ronneby, Sweden" (PMEDICINE-D-25-01709R1) to PLOS Medicine. The paper has been reviewed by subject experts and a statistician; their comments are included below and can also be accessed here: [LINK]

As you will see, the reviewers have found your study of considerable interest but have raised some important technical concerns that need to be resolved in full before resubmission. After discussing the paper with the editorial team, I'm pleased to invite you to revise the paper in response to the reviewers' comments. We plan to send the revised paper to some or all of the original reviewers, and we cannot provide any guarantees at this stage regarding publication. Please be advised that we may invite a third independent reviewer to consider a revised manuscript.

We ask that you submit your revision by Dec 15 2025 11:59PM. However, if this deadline is not feasible, please contact me by email, and we can discuss a suitable alternative.

Don't hesitate to contact me directly with any questions (acunha@plos.org).

Best regards,

Andreia

Andreia Cunha, PhD

Senior editor

PLOS Medicine

acunha@plos.org

Comments from the reviewers:

Reviewer #1: Blomberg and team present an important study investigating antenatal exposure to PFAS and childhood asthma. They have utilised a unique Swedish cohort who were exposed to high levels of PFAS in drinking water over several years and using linkage to Swedish registers were able to investigate offspring outcomes. The manuscript is well written and the analysis appropriate.

Main queries:

1. It is unclear why the main analysis is not that using the imputed dataset and then accounting for clustering using this dataset. This is the more robust analysis and given there are siblings within the cohort the independence assumption of the model is violated in their current complete case primary analysis and should be avoided.

2. How was the independence assumption handled in the unadjusted Kaplan-Meier analysis?

3. Was there an a-priori statistical analysis plan? If so, this should be included in the supplemental files.

4. Further details of the causal analysis are needed, was there a trial target framework established a-priori or consideration of casual assumptions. Were the imputed data used for this analysis?

5. In the causal matched analysis, were the models adjusted for post matching? This is recommended to account for residual imbalance and doubly robust estimation.

6. Line 327 – the unadjusted HR has a lower CI of 1.00 – suggesting no association, the authors should not overstate an association of unadjusted results.

7. It is difficult to discern what is antennal exposure vs early childhood exposure in this population. The authors have mentioned this in the limitations. However, I am wondering whether they have data on how many stayed within the exposure area post birth and during the period of water contamination?

8. Clarification of the outcome ascertainment around wheeze is needed, it’s unclear why J45 is included in this outcome and may explain the low sensitivity.

Minor comments:

1. The use of sharp null hypothesis should be explained; this will not be a concept familiar to most readers.

2. The validation of the exposure is a significant strength of the study and analysis appropriate. Given the samples were collected from 2014 – 2016, were there differences between those collected remote from exposure which may have influenced levels?

3. The higher level of exposure in the present study should be a greater focus in the discussion of previous studies as this has likely contributed to the differences.

4. Discussion of the applicability of the present study to the general population, given such very high levels are likely only rarely seen.

5. The use of maternal address as a proxy should be noted as a limitation.

6. Parental asthma will not capture childhood asthma among parents this should be noted as a limitation.

7. Supplemental figures should include point estimates and 95% CI so they can be compared with main analysis

Reviewer #2: Thank you for the opportunity to review this manuscript. The authors examined the relationship between prenatal PFAS exposure and wheezing and asthma in Ronneby. This study is noteworthy, as similar research conducted in highly exposed populations is rare. The manuscript is well-supported by preceding validation studies and is clearly written. I have only several minor concerns remaining.

1. Line 84: Please consider using a flow chart to display the participant selection process.

2. Line 130: Given that the study period spans 2006 to 2013 while the validation study for exposure assessment was conducted from 2014 to 2016, please consider performing a sensitivity analysis that restricts the study period to the final years or latter half, where the validation study findings are more likely to be applicable.

3. Line 138: Please provide a brief description of the Swedish medical system, although this point is addressed in the study limitations section.

4. Line 184: The authors appear to use two different methods to indicate asthma presence (true/false or yes/no). Please clarify this inconsistency.

5. Line 202: Please provide details of the procedures used to test the proportional hazard assumptions.

6. Line 435: The authors stated that children in the high exposure group continued to experience postnatal exposure. This assertion appears inaccurate for children born during the 2010s. Did area residents continue consuming contaminated water subsequent to 2013?

---

* Please upload any figures associated with your paper as individual TIF or EPS files with 300dpi resolution at resubmission; please read our figure guidelines for more information on our requirements: http://journals.plos.org/plosmedicine/s/figures. While revising your submission, we strongly recommend that you use PLOS's NAAS tool (https://ngplosjournals.pagemajik.ai/artanalysis) to test your figure files. NAAS can convert your figure files to the TIFF file type and meet basic requirements (such as print size, resolution), or provide you with a report on issues that do not meet our requirements and that NAAS cannot fix.

After uploading your figures to PLOS's NAAS tool - https://ngplosjournals.pagemajik.ai/artanalysis, NAAS will process the files provided and display the results in the "Uploaded Files" section of the page as the processing is complete.

If the uploaded figures meet our requirements (or NAAS is able to fix the files to meet our requirements), the figure will be marked as "fixed" above. If NAAS is unable to fix the files, a red "failed" label will appear above.

When NAAS has confirmed that the figure files meet our requirements, please download the file via the download option, and include these NAAS processed figure files when submitting your revised manuscript.

FIGURES AND TABLES

SUPPLEMENTARY MATERIAL

REFERENCES

OBSERVATIONAL STUDIES

* Abstract: Please include the study design, population and setting, number of participants, years during which the study took place (enrollment and follow up), length of follow up, and main outcome measures.

* Please ensure that the study is reported according to the STROBE (or appropriate STOBE extension) guideline (available from: https://www.equator-network.org/reporting-guidelines/strobe) and include the completed STROBE (or STROBE extension) checklist as Supporting Information. Please add the following statement, or similar, to the Methods: "This study is reported as per the Strengthening the Reporting of Observational Studies in Epidemiology (STROBE) guideline (S1 Checklist)." When completing the checklist, please use section and paragraph numbers, rather than page numbers.

* If more appropriate, please ensure that the study is reported according to the RECORD guideline (available from https://www.record-statement.org) and include the completed checklist as Supporting Information. Please add the following statement, or similar, to the Methods: "This study is reported as per the Reporting of Studies Conducted using Observational Routinely-Collected Data (RECORD) guideline (S1 Checklist)." When completing the checklist, please use section and paragraph numbers, rather than page numbers.

* For all observational studies, in the manuscript text, please indicate: (1) the specific hypotheses you intended to test, (2) the analytical methods by which you planned to test them, (3) the analyses you actually performed, and (4) when reported analyses differ from those that were planned, transparent explanations for differences that affect the reliability of the study's results. If a reported analysis was performed based on an interesting but unanticipated pattern in the data, please be clear that the analysis was data driven.

* Please state in the Methods section whether the study had a prospective protocol or analysis plan. If a prospective analysis plan (from your funding proposal, IRB or other ethics committee submission, study protocol, or other planning document written before analyzing the data) was used in designing the study, please include the relevant document(s) with your revised manuscript as a Supporting Information file to be published alongside your study and cite it in the Methods section. A legend for this file should be included at the end of your manuscript. If no such document exists, please make sure that the Methods section transparently describes when analyses were planned, and when/why any data-driven changes to analyses took place. Changes in the analysis, including those made in response to peer review comments, should be identified as such in the Methods section of the paper, with rationale.

---

## [Decision Letter · Decision Letter 2]

30 Jan 2026

Dear Dr. Blomberg,

Sincere apologies for the delay in getting back to you with a decision which was due to challenges in securing all the necessary advice. Thank you very much for re-submitting your manuscript "Prenatal exposure to perfluoroalkyl substances (PFAS) and incidence of asthma and wheeze in childhood: A cohort study in Ronneby, Sweden" (PMEDICINE-D-25-01709R2) for review by PLOS Medicine.

I have discussed the paper with my colleagues and the academic editor and it was also seen again by the two original reviewers and by a third reviewer. I am pleased to say that provided the remaining reviewer 3, editorial and production issues are dealt with we are planning to accept the paper for publication in the journal.

[LINK]

We look forward to receiving the revised manuscript by Feb 06 2026 11:59PM.

Sincerely,

Andreia Cunha, PhD

Senior Editor

PLOS Medicine

plosmedicine.org

Requests from Editors:

GENERAL EDITORIAL REQUESTS

* Please confirm that your title complies with PLOS Medicine's style. Your title must be nondeclarative and not a question. It should begin with main concept if possible. "Effect of" should be used only if causality can be inferred, i.e., for an RCT. Please place the study design ("A randomized controlled trial," "A retrospective study," "A modelling study," etc.) in the subtitle (ie, after a colon).

* Please confirm that your abstract complies with our requirements, including format (three sections: Background, Methods and Findings, and Conclusions) and providing all the information relevant to this study type https://journals.plos.org/plosmedicine/s/submission-guidelines#loc-abstract

* Please ensure that the Introduction ends with a clear description of the study question or hypothesis.

* Please ensure that all abbreviations are defined at first use throughout the text.

* Please confirm that all numbers presented in the abstract are present and identical to numbers presented in the main manuscript text.

GENERAL

* Please review your text for claims of novelty or primacy (e.g. 'for the first time') and remove this language. In addition, please check that any use of statistical terms (such as trend or significant) are supported by the data, and if not please remove them.

* Please remove the 'conclusions' subheading from the discussion. Please also remove any other subheadings from the discussion.

* Statistical reporting: Please revise throughout the manuscript, including tables and figures.

- Please report statistical information as follows to improve clarity for the reader (95% CI [13,28]; p</=).

- Please separate upper and lower bounds with commas instead of hyphens as the latter can be confused with reporting of negative values.

- Please repeat statistical definitions (HR, CI etc.) for each set of parentheses."

* In the abstract, please include the important dependent variables that are adjusted for in the analyses.

* Please replace "subject" with participant, patient, individual, or person.

FUNDING STATEMENT

* The funding statement should include: specific grant numbers, initials of authors who received each award, URLs to sponsors’ websites. Also, please state whether any sponsors or funders (other than the named authors) played any role in study design, data collection and analysis, the decision to publish, or preparation of the manuscript. If they had no role in the research, include this sentence: “The funders had no role in study design, data collection and analysis, decision to publish, or preparation of the manuscript.”

COMPETING INTERESTS STATEMENT

* All authors must declare their relevant competing interests per the PLOS policy, which can be seen here: https://journals.plos.org/plosmedicine/s/competing-interests For authors with ties to industry, please indicate whether any of the interests has a financial stake in the results of the current study.

DATA AVAILABILITY

* PLOS Medicine requires that the de-identified data underlying the specific results in a published article be made available, without restrictions on access, in a public repository or as Supporting Information at the time of article publication, provided it is legal and ethical to do so. Please see the policy at

http://journals.plos.org/plosmedicine/s/data-availability

and FAQs at

http://journals.plos.org/plosmedicine/s/data-availability#loc-faqs-for-data-policy "

* The Data Availability Statement (DAS) requires revision. For each data source used in your study:

ETHICS AND CONSENT

* Please specify the reason informed consent was not required. Please ensure that the research complies with the PLOS policy in full: https://journals.plos.org/plosmedicine/s/human-subjects-research#loc-patient-privacy-and-informed-consent-for-publication

FIGURES

* Please define all elements of box plots in the figure caption - center line, box limits and whiskers.

* Please provide titles and legends for all figures and tables (including those in Supporting Information files). Please define all acronyms used in each figure or table in its corresponding legend.

* Please ensure that where relevant figures include 95% CIs.

OBSERVATIONAL, COHORT, CROSS-SECTIONAL, AND CASE CONTROL STUDIES

* Please ensure that the study is reported according to the STROBE guideline, and include the completed STROBE checklist as Supporting Information. Please add the following statement, or similar, to the Methods: ""This study is reported as per the Strengthening the Reporting of Observational Studies in Epidemiology (STROBE) guideline (S1 Checklist).

* Did your study have a prospective protocol or analysis plan? Please state this (either way) early in the Methods section.

c) In either case, changes in the analysis-- including those made in response to peer review comments-- should be identified as such in the Methods section of the paper, with rationale."

* Your study is observational and therefore causality cannot be inferred. Please remove language that implies causality and refer to associations instead.

* For all observational studies, in the manuscript text, please indicate: (1) the specific hypotheses you intended to test, (2) the analytical methods by which you planned to test them, (3) the analyses you actually performed, and (4) when reported analyses differ from those that were planned, transparent explanations for differences that affect the reliability of the study's results. If a reported analysis was performed based on an interesting but unanticipated pattern in the data, please be clear that the analysis was data-driven.

* If more appropriate than STROBE, please ensure that the study is reported according to the RECORD guideline (available from https://www.record-statement.org) and include the completed checklist as Supporting Information. Please add the following statement, or similar, to the Methods: "This study is reported as per the Reporting of Studies Conducted using Observational Routinely-Collected Data (RECORD) guideline (S1 Checklist)." When completing the checklist, please use section and paragraph numbers, rather than page numbers

Comments from Reviewers:

Reviewer #1: I thank the authors for their considered replies and update of the manuscript. I have no further comments.

Reviewer #2: Thank you for your consideration. I have no further comments at this time.

Reviewer #3: This manuscript presents a well-conducted register-based cohort study investigating the association between prenatal exposure to PFAS from contaminated drinking water and the incidence of asthma and wheeze in children. Leveraging a unique natural experiment in Ronneby, Sweden, where a subset of the population was exposed to very high PFAS levels, the study addresses an important gap in the literature, as prior epidemiological studies have largely focused on background exposure levels. The key finding is a statistically significant increased risk of asthma (but not early-childhood wheeze) among children with very high prenatal PFAS exposure. The study employs robust methods, including careful exposure categorization, validation of outcomes, Cox proportional hazards models, and a secondary Rubin Causal Model analysis. The conclusions are appropriately cautious, highlighting the specificity of the findings to very high exposure levels.

Abstract: One minor suggestion: in the "Methods and Findings" summary, consider specifying that the increased asthma risk was observed specifically in the very high exposure group, as this is a crucial nuance.

Introduction: The introduction effectively establishes the public health significance of childhood asthma, the environmental concern posed by PFAS, and the specific knowledge gap this study aims to fill. The rationale for studying a highly exposed population is compelling. To further strengthen it, you might briefly mention the mixed evidence from prior longitudinal studies on prenatal PFAS exposure and asthma, which would provide a clearer segue into your study's novel contribution.

Methods: Exposure Assessment: In the main text or supplement, consider adding a brief discussion of the potential implications of using pre-delivery maternal address as a proxy for fetal exposure, acknowledging it as a reasonable but imperfect measure.

Results: When presenting the sex-stratified results in the text or supplement, a brief interpretation of the marginal interaction for wheeze (p=0.08) and the direction of effects could be helpful, even if not statistically definitive.

Discussion: (1) Strengths Section: You may consider adding the large sample size and the use of time-to-event analysis as explicit strengths. (2) Limitations: The discussion of exposure misclassification is excellent and appropriately nuanced (differentiating error structures). The point about ongoing childhood exposure for many children is critical and is well-handled. You might briefly reiterate here that this precludes attributing effects solely to the prenatal period.

Conclusions: The conclusions are balanced and justified. Consider softening the statement "may not generalize to populations with lower exposures" to "likely have limited generalizability to populations with only background-level exposures," as this aligns with your threshold-effect hypothesis.

[LINK]

---

## [Editor Report · Decision Letter 3]

23 Feb 2026

Dear Dr. Blomberg,

Thank you very much for re-submitting your manuscript "Prenatal exposure to perfluoroalkyl substances (PFAS) and incidence of asthma and wheeze in childhood: A cohort study in Ronneby, Sweden" (PMEDICINE-D-25-01709R3) for review by PLOS Medicine.

We have now carefully assessed the revisions in your manuscript and there are a few remaining editorial points that need addressing before we can proceed to accept the paper for publication in the journal.

* Please ensure the Abstract is proofread, no sentence starts with a number and the acronym AFFF is defined on first use.

* Please add the limitations of the study as the last point in the author summary.

*The funding statement should include the URLs to sponsors’ websites rather than hyperlinks.

* Please clarify if the data was anonymized and whether this was the reason informed consent was not required “The Swedish Ethical Review Authority approved the study (number 2021-04872) and waived the requirement for individual informed consent because the research used existing registry data and could not practicably be conducted if consent were required.”

* Please remove any mention of suggestive evidence if not statistically significant and simply describe the results. We appreciate this is not in line with the suggestion from Reviewer 4 but it is a journal policy requirement.

* Since PFAS exposure after the pre-natal period cannot be ruled out, please consider if using early-life exposure instead (or another more encompassing term) in the title and text might be more accurate.

[LINK]

In revising the manuscript for further consideration here, please ensure you address the specific points made by the editors. In your rebuttal letter you should indicate your response to the editors' comments and the changes you have made in the manuscript. Please submit a clean version of the paper as the main article file. A version with changes marked must also be uploaded as a marked up manuscript file.

We look forward to receiving the revised manuscript by Mar 05 2026 11:59PM.

Sincerely,

Andreia Cunha, PhD

Senior Editor

PLOS Medicine

plosmedicine.org

[LINK]

---

## [Editor Report · Decision Letter 4]

3 Mar 2026

Dear Dr Blomberg,

On behalf of my colleagues and the Academic Editor, Dr Sanjay Basu, I am pleased to inform you that we have agreed to publish your manuscript "Prenatal exposure to perfluoroalkyl substances (PFAS) and incidence of asthma and wheeze in childhood: A cohort study in Ronneby, Sweden" (PMEDICINE-D-25-01709R4) in PLOS Medicine.

Please update the title to comply with PLOS Medicine's guidelines to read: Prenatal exposure to perfluoroalkyl substances and incidence of asthma and wheeze in childhood: A register-based cohort study in Ronneby, Sweden.

Please also ensure that throughout the text you either use highly-exposed or high-exposure population but not high-exposed.

PRESS

Sincerely,

Andreia Cunha, PhD

Senior Editor

PLOS Medicine